# Impact of Deprivation on Obesity in Children with PWS

**DOI:** 10.3390/jcm11082255

**Published:** 2022-04-18

**Authors:** Sabrina Grolleau, Marine Delagrange, Melina Souquiere, Catherine Molinas, Gwenaëlle Diene, Marion Valette, Maithé Tauber

**Affiliations:** 1French Reference Centre for Prader-Willi Syndrome and Other Rare Forms of Obesity with Eating Disorders, CHU Toulouse, 31059 Toulouse, France; grolleau.s@chu-toulouse.fr (S.G.); delagrange.m@chu-toulouse.fr (M.D.); souquiere.me@chu-toulouse.fr (M.S.); molinas.c@chu-toulouse.fr (C.M.); diene.g@chu-toulouse.fr (G.D.); valette.m@chu-toulouse.fr (M.V.); 2UMR 1027 INSERM, University Toulouse III, 31062 Toulouse, France; 3UMR1291 INSERM-CNRS-UMR5051, University Toulouse III, 31062 Toulouse, France

**Keywords:** Prader–Willi syndrome, obesity, socioeconomic status, deprivation

## Abstract

Our study aimed to evaluate the social deprivation score in families with a child with Prader-Willi syndrome (PWS) and analyze its impact on the occurrence of obesity in the affected child. We included 147 children with PWS followed in our reference center with Evaluation of the Deprivation and Inequalities of Health in Healthcare Centres by the EPICES score. Deprivation (EPICES ≥ 30) was found in 25.9% of the population. Compared with the non-obese children, children with obesity had more deprived families, 50.0 vs. 18.0% (*p* = 0.0001); were older, with a median of 10.1 vs. 6.0 years (*p* = 0.0006); were less frequently treated with growth hormone (GH), 80.6 vs. 91.9% (*p* = 0.07). The mothers of obese children were more frequently obese, 46.9 vs. 13.3% (*p* < 0.0001), and achieved high study levels less frequently (≥Bac+2), 40.9 vs. 70.1% (*p* = 0.012). The multivariate logistic regression indicated that age, living in a deprived family, and having a mother with overweight/obesity were significantly associated with an increased risk of obesity (respectively, OR = 3.31 (1.26–8.73) and OR = 6.76 (2.36–19.37)). The same risk factors of obesity observed in the general population were found in children with PWS. Families at risk, including social deprivation, will require early identification and a reinforced approach to prevent obesity.

## 1. Introduction

Prader–Willi syndrome (PWS) is a rare genetic neurodevelopmental disorder with an incidence of 1/21,000 births [1] due to the loss of paternally inherited imprinted genes at 15q11.2-q13. In about half of the cases, the cause is a paternal 15q11-q13 deletion (DEL) and, in the remaining cases, the cause is a maternal uniparental disomy of chromosome 15 (mUDP 45%) or, more rarely, an imprinting defect or a translocation. The phenotype is related to hypothalamic dysfunction [2]. PWS is characterized by a unique developmental, nutritional, endocrine, metabolic, and behavioral trajectory over a lifetime. The nutritional phases well described by the team in Florida comprises six post-natal phases [3]. In the neonatal period, infants display severe hypotonia, and sucking and swallowing deficits that impairs normal feeding and weight gain. Around 80% of neonates require nasogastric tube feeding to prevent failure to thrive. Then, young children develop excessive weight gain, hyperphagia, and obsession with food and food-related issues that leads to severe early obesity in the absence of strict control of food access in a firm and caring environment. Patients have impaired body composition with lower muscle mass and higher fat mass than in the general population with comparable body mass index (BMI) [4,5], which explains at least in part decreased resting energy expenditure [5,6]. Endocrine dysfunction comprises growth hormone (GH) deficiency (most children and >80%), hypogonadism of mixed origin (90% of patients), and central hypothyroidism (20–40% of patients) [2,7,8]. A few cases of adrenal insufficiency have also been described, but the prevalence appears to be low [9,10]. Most patients have intellectual disability with moderate or light cognitive deficit, learning disability, poor social skills and regulation of emotion, and behavioral disorders—in some cases, severe psychiatric disorders. We published a psychopathological pattern of the disease, classifying patients into basic, impulsive, compulsive, and psychotic patterns [11]. More recently, we proposed a classification of symptoms according to the Research Domain Criteria (RDoC) [12] and proposed that hyperphagia observed in PWS is closed to addiction for food and food-related issues [13]. GH supplementation is now started during the first months of life soon after diagnosis and implementation of multidisciplinary care. Its positive effects on body composition, growth, metabolic status, and adult height have been demonstrated in several studies [14,15,16,17]. Besides early diagnosis and implementation of multidisciplinary care including GH treatment, we observed that some children and among them those from deprived families still develop early onset of obesity.

The EPICES score (Évaluation de la précarité et des inégalités de santé dans les centres d’examens de santé, Evaluation of the Deprivation and Inequalities of Health in Healthcare Centres) is a validated deprivation score in France [18] considering multiple dimensions of socioeconomic conditions, including psychological aspects. Social deprivation is defined by limited access to society’s resources due to poverty, discrimination, or other disadvantages. In this study we want to evaluate the social deprivation score in families with a child with PWS and analyze its impact on the occurrence of obesity in the affected child.. We also want to determine predictors of obesity in children with PWS including the EPICES score. We also want to determine predictors of obesity in children with PWS, particularly with respect to SES.

## 2. Materials and Methods

We conducted a monocentric descriptive study on patients with PWS followed at least for one year in the reference center with expertise in PWS and obesity for more than 15 years. Families were admitted for 2 days and were seen by a multidisciplinary team comprising the pediatric endocrinologist with expertise on obesity and PWS, the dietician, the psychologist and the psychiatrist, the speech and language expert, ENT specialist, sleep expert and orthopedist with other professionals when needed. A letter summarizing the evaluations and sampling performed was sent to the general practitioner and to the local health professionals. No medical treatment for obesity was used. Growth hormone treatment was started in all children at the age of 6 months. Other hormonal substitutions were performed at all ages based on hormonal levels and vitamin D was given systematically. The scheduled visits of PWS patients were organized depending on the age of the children every 3 months in infants and every 6 months later young children. From October 2016 to June 2018, all families coming for a scheduled visit to the reference center were included and were asked to fill the EPICES questionnaire. This questionnaire is a 11 items questionnaire (see Appendix A) which evaluates the deprivation of the family and was validated in a large cohort of 197 389 persons in France. The higher the score, the more deprived the family is; a cutoff of 30 was used to define deprivation. The pediatrician explained and gave the questionnaire to the family. All families accepted to participate to this study and were non-opposed to the use of their child data for statistical analysis. Our team of the reference centre collected clinical and socio-demographics information of the children and parents: for the children, height, weight were measured during the routine visit in our department. Medications (GH treatment, psychotropic treatment, non-invasive ventilation (NIV)…) were collected. At this visit, some of the patients filled the Hyperphagia questionnaire (HQ) a specific questionnaire developed for PWS [19] converted to HQ for clinical trials HQ-CT [20]; this HQ was not filled at each visit in routine care and not all the patients included in the study filled it.

Parent’s data were collected during the physician interview with the parents: height and weight were declared and not measured. Both parents were well-known by the caregivers of our department. We collect informations from parents regarding BMI and education levels. 

All these data are recorded in our national database of the PWS reference centre with was approved by the CNIL in France. No submission to the ethics committee was needed.

### 2.1. Definition of obesity

Children obesity was defined as World Health Organization (WHO) BMI Z-score greater than +2. In adult overweight and obesity were defined by a BMI >25 kg/m2 and >30 respectively [21,22].

### 2.2. Statistical Analysis

All statistical analysis were performed using SAS^®^, version 9.4 (SAS^®^, Cary, NC, USA). Descriptive statistics were used for the demographic characteristics of the sample. Proportions were compared using the approximate **χ^2^** test, or Fisher’s exact test when necessary. The t-test was applied when groups were compared in terms of continuous variables provided that they were fairly normally distributed. If the variable was not normally distributed, the Wilcoxon test was used. A *p*-value of less than 0.05 was considered statistically significant.

Characteristics independently associated with obesity were explored using a model of multivariate logistic regression. Adjusted odd ratios were shown with their 95% confidence intervals.

Statistical graphics using LOcally Estimated Scatterplot Smoothing (LOESS) curves were used to evaluate the relation between BMI by age of children. The LOESS curves use non-parametric methods. The term “LOESS” is an acronym for “local regression” and the entire procedure is a fairly direct generalization of traditional least-squares methods for data analysis [23].

## 3. Results

Study population at inclusion and prevalence of deprivation:

We included 147 PWS families. Table 1 describes the characteristics of the whole population at inclusion in the study and in the two groups of children with or without obesity. An EPICES score ≥ 30 defining deprivation was found in 25.9% of the families, which is lower than the prevalence in the French general population 35% (18) and in children with obesity (around 60%) [24].

The median age of the population was 7.1 years, and the mean age was 8.1 ± 5.2 SD (children aged from 3 months to 19 years). Compared with the non-obese children, children with obesity more often had a deprived family, 50.0 vs. 18.0% (*p* = 0.0001) (Figure 1); were older, 10.1 vs. 6.0 years in median (*p* = 0.0006); had a trend for being less frequently treated with GH, 80.6 vs. 91.9% (*p* = 0.069); and used NIV more frequently 22.2 vs. 8.1% (*p* = 0.033). Albeit the mean score of HQ seems to be higher in obese children (mean HQ score 25.3 ± 7.2 vs. 20.4 ± 7.5, *p* = 0.22, and HQ-CT score 9.7 ± 4.5 vs. 6.3 ± 6.0, *p* = 0.19), any statistically difference was observed probably due to the low number of data available. Children with obesity were more frequently educated in an adapted school, 88.9 vs. 76.8%, but this difference was not statistically confirmed (*p* = 0.19). Among the obese children, the mothers displayed overweight/obesity more frequently, 46.9 vs. 13.3% (*p* < 0.0001), and achieved high study levels less frequently (≥Bac+2), 40.9 vs. 70.1% (*p* = 0.012). No differences were observed in the obesity and level of study of fathers.

### 3.1. Comparison of the Characteristics of Children with and without Deprivation

Table 1 and Figure 2 show the characteristics of children in the deprived and non-deprived groups with data of the whole population for better overview. The prevalence of obesity was higher in the deprived group, 47.4 vs. 18.4% (*p* = 0.0004). The mothers were more frequently overweight/obese in the deprived group vs. the non-deprived group (35.3 vs. 16.7%, *p* = 0.023). Mothers and fathers in the deprived families achieved a less high study level (respectively, 26.1 vs. 75.0%, *p* < 0.0001, and 36.4 vs. 67.1%, *p* = 0.01). Although there was no significant difference on the age at diagnosis between the two groups, there was a wider range of mean age at diagnosis with 7.7 ± 15.6 months in the deprived group vs. 1.8 ± 3.2 months in the non-deprived group.

### 3.2. Comparison of the Characteristics of the Children with or without Obesity in the Deprived Group

Table 2 shows the characteristics of children with a deprivation level classified by obesity. In the deprived group, 18 children (47.4%) were obese and 20 (52.6%) were lean. The only significant difference between the two groups were age, with higher mean age at inclusion 11.1 ± 5.0 vs. 7.1 ± 4.9 years (*p* = 0.021) and higher mean age at diagnosis 9.6 ± 16.7 (median 1.0 (Q1:1.0–Q3:12.0)) vs. 5.9 ± 14.7 months (median 0.7 (Q1:0.4–Q3:1.3)) (*p* = 0.024) in the children with obesity. There was a trend for the mothers of the children with obesity to display more frequently overweight/obesity, 50.0 vs. 22.2% (*p* = 0.091).

### 3.3. Multisteps Analysis

Table 3 shows the multivariable logistic regression analysis of obesity in children adjusted on variables models (N = 130). The multivariate logistic regression analysis indicated that age (OR = 1.16 for each additional year IC95% (1.06–1.28)), having a family in a situation of deprivation, and having a mother with overweight/obesity were significantly associated with an increased risk of obesity in children (respectively, OR = 3.31 (1.26–8.73) and OR = 6.76 (2.36–19.37)). Moreover, the univariate logistic regression analysis showed that the mother’s high level of education (≥Bac+2) significantly reduced the risk of child obesity (OR = 0.29 (0.11–0.79), *p* = 0.015), but these data was not included in the multivariate model due to a high amount of missing data.

## 4. Discussion

We showed for the first time in PWS that obesity is more prevalent in children from deprived families (47% vs. 18%). The EPICES score may be a useful tool to routinely identify families who may benefit from additional support for care during follow-up, particularly to prevent the occurrence of early obesity. In addition, we identified other factors associated with obesity. The age of the child is associated with obesity, and this is explained by the known trajectory of the disease starting from difficulties in sucking and gaining weight to hyperphagia and obesity. Interestingly we were able to show that education level of the mother and the presence of mother overweight/obesity are associated with obesity, independently in both deprived and non-deprived families. Therefore, early identification of vulnerable families should be implemented as soon as possible after the diagnosis of PWS, which is made in the first months of life in order to mitigate their impact on the occurrence of obesity.

A recent review [25] examined the socio-ecological model of obesity in children described in the scientific literature and found that the strong and inverse relationship between obesity and socioeconomic status in high-income countries can be explained by different mechanisms including psychosocial problems. Several studies have found an association between psychosocial well-being and overweight [26,27]. One of the factors that seems to mediate all these relationships is stress derived from these most unfavorable situations, and various articles have linked the stress maintained over time with obesity and metabolic syndrome [25].

In children displaying obesity without PWS, most of the prevention programs identified did not consider social vulnerabilities and inequalities making them ineffective in most vulnerable groups.

Interventions conducted in children from socially vulnerable group suggest modest but promising effects [28]. Current evidence suggests that prevention programs should include a combination of policies that address structural barriers (at the school, community, or society level) and strategies involving the child and his family [25].

The burden of the family having a child with PWS is strong after diagnosis and remained elevated throughout life [29]. This stress could be involved in the occurrence of obesity and particularly in unfavorable psychosocial situations.

Interestingly in our study we could demonstrated that albeit prevalence of obesity is higher in children with PWS in deprived families, half of them benefit from the early multidisciplinary care and were not obese. It has been well documented that obesity is a constant feature in children with PWS when early diagnosis and a multidisciplinary approach is not implemented. Obesity is highly prevalent in deprived population. Therefore, we could hypothesize that obesity may be present in all children with PWS from deprived families. Nevertheless, our findings documented that half of the children in deprived families were not obese suggesting that the multidisciplinary approach and the follow-up by an expert centre in links with the general practitioner or local paediatrician have positive outcome. Unfortunately, we did not find publication on the impact of deprivation on the occurrence of obesity in children with PWS. This is a strong signal showing that in addition to a disease-focused approach, a family-centered approach for obesity may result in a better outcome. Therefore, strong efforts are needed to identify deprived families early on in order to adjust our support and care.

### Limitations and Strengths of the Study

There are many limitations of our study. The first one is the small number of patients with a wide age range that prevents separating into groups according to age. Another limitation of our study is that it has been conducted in a single center. However, being a national reference center, we admit families from all over the country that might reduce the center effect. In addition, we also organize regular training, formation and case discussions for professionals wanting to discuss of complex situations. Another limitation is that we do not have the age of the parents and the history of obesity of the parents and various missing data regarding the hyperphagia questionnaire which is not done routinely and other data about parent’s level of education. The strength of the study is that families received early diagnosis and care for PWS based on a national protocol named PNDS (protocole national de diagnostic et de soins) which is a national protocol for diagnosis and care of PWS. The protocol is written by national experts with multidisciplinary dimensions and is published and regularly updated by the Haute Autorité de Santé (HAS), the institution that organizes and broadcasts via internet the PNDS for rare diseases. Health professionals may read the protocol and have a view of the optimal care for PWS.

## 5. Conclusions

Prevention of obesity in children with PWS remains a priority target for action. Early identification of vulnerable families is required as the prevalence of obesity in children with PWS from deprived families is higher compared to non-deprived families. The use of the EPICES score is helpful and shows that not all children with PWS of vulnerable families will become obese, demonstrating that the early diagnosis and multidisciplinary care have some effects but are insufficient. Families at risk, including social deprivation, will require early identification and a reinforced approach to prevent obesity.

## Figures and Tables

**Figure 1 jcm-11-02255-f001:**
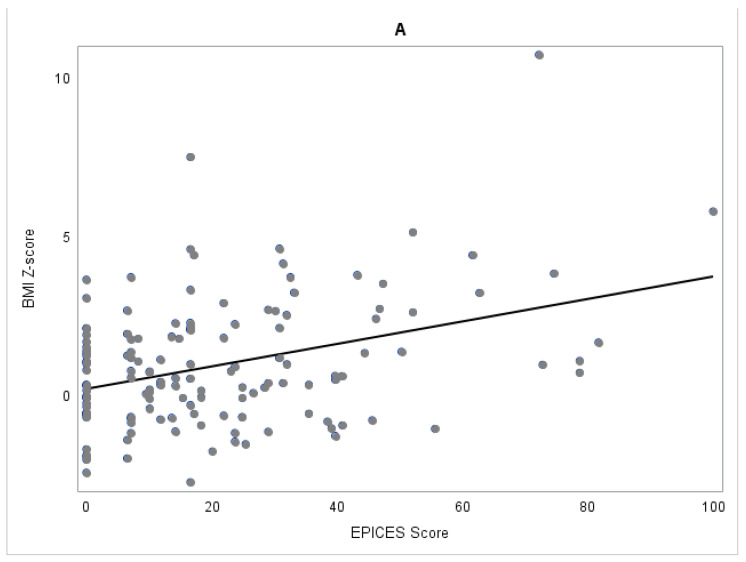
(**A**) Scatter plot and regression line between EPICES score (*X*-axis) and BMI Z-score (*Y*-axis) in the 147 children with PWS. (**B**) Boxplot of EPICES score in the obese group vs. the non-obese group.

**Figure 2 jcm-11-02255-f002:**
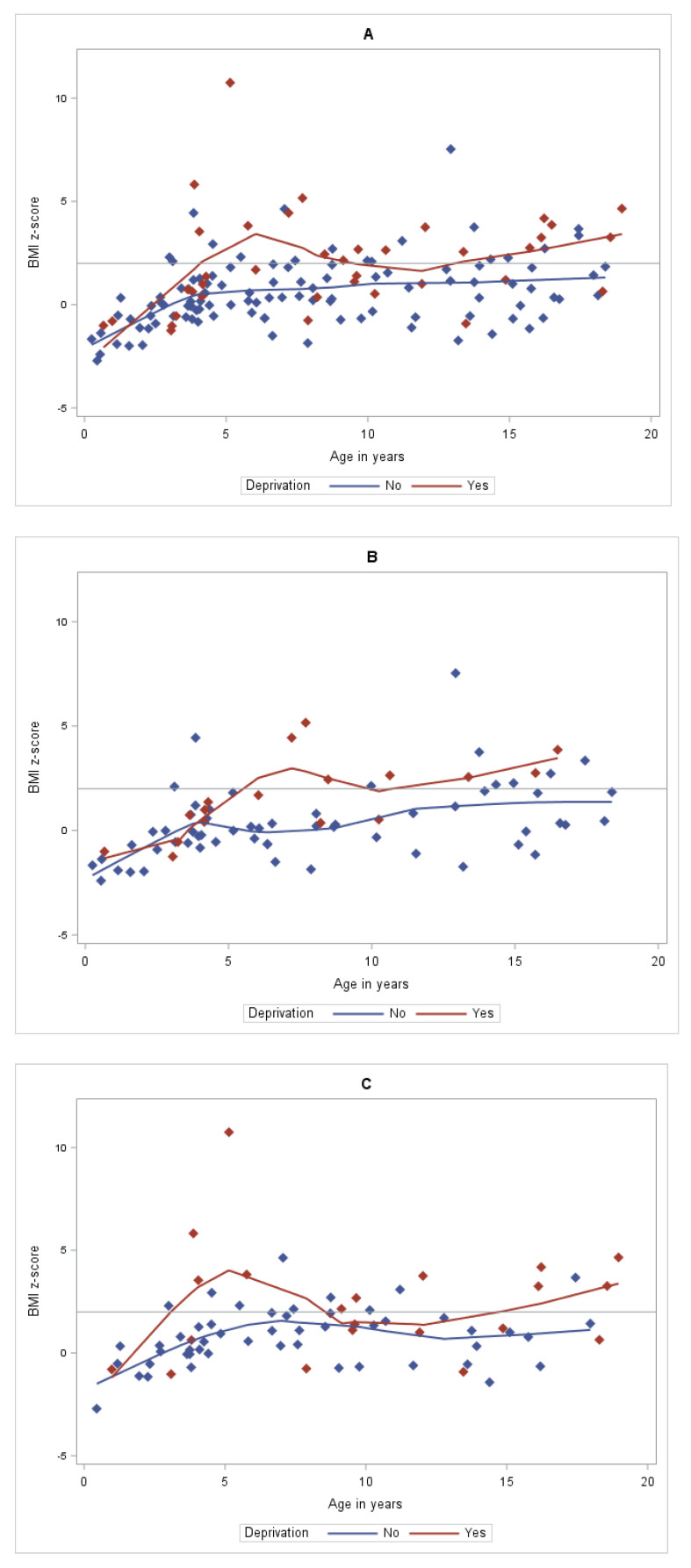
Scatter plot and Loess curves of BMI by age in years of children and deprivation (**A**) in the 147 children with PWS; (**B**) in the 75 boys with PWS; (**C**) in the 72 girls with PWS. In blue, non-deprived patients, and in red, deprived patients.

**Table 1 jcm-11-02255-t001:** Characteristics of children and their parents in the whole group and in the four groups, obese vs. non-obese and deprived vs. non-deprived. Variables are expressed in *n* (%) or in Med/Mean ± SD; for continuous variables (age, age at diagnosis, BMI Z-score, HQ, HQ-CT) P was calculated using Student or Wilcoxon test; for categorical variables (gender, genetic diagnosis, special schooling, parent’s obesity, study level of parents, treatment of GH/Psychotropic treatment, NIV, deprivation) *p* was calculated using Chi2 or Fisher exact test.

	All Patients N = 147	Non-Obese N = 111 (75.5%)	Obese * N = 36 (24.5%)	*p*	Non Deprivation N = 109 (74.1%)	Deprivation ** N = 38 (25.9%)	*p*
Boys	75 (51.0)	59 (53.2)	16 (44.4)	0.36	58 (53.2)	17 (44.7)	0.37
Age in years ***	7.1/8.1 ± 5.2	6.0/7.3 ± 5.0	10.1/10.6 ± 5.0	**0.0006**	6.6/7.8 ± 5.1	8.3/9.0 ± 5.3	0.22
Age at diagnosis in months	1.0/3.3 ± 8.7	1.0/2.7 ± 7.0	1.0/5.2 ± 12.2	0.16	1.0 (1.8 ± 3.2)	1.0/7.7 ± 15.6	0.13
Genetic subtype Deletion (*n* = 146)	71 (48.6)	52 (46.9)	19 (54.3)	0.44	53 (49.1)	18 (47.4)	0.86
Obesity (BMI z-score > 2)	36 (24.5)	-	-	-	18 (16.5)	18 (47.4)	**0.0001**
GH treatment of children	131 (89.1)	102 (91.9)	29 (80.6)	0.069	100 (91.7)	31 (81.6)	0.13
Psychotropic treatment	16 (10.9)	11 (9.9)	5 (13.9)	0.54	11 (10.1)	5 (13.2)	0.56
NIV	17 (11.6)	9 (8.1)	8 (22.2)	**0.033**	12 (11.0)	5 (13.2)	0.77
HQ Total Score (*n* = 40)	19.5/20.8 ± 7.5	19.0/20.4 ± 7.5	29.0/25.3 ± 7.2	NA ^£^	20.0/20.6 ± 7.6	18.0/21.6 ± 7.3	NA ^µ^
HQ-CT (*n* = 40)	4.5/6.6 ± 6.0	4.0/6.3 ± 6.0	10.0/9.7 ± 4.5	NA ^£^	5.0/6.6 ± 6.1	4.0/6.2 ± 5.6	NA ^µ^
Special schooling (*n* = 83)	67 (80.7)	43 (76.8)	24 (88.9)	0.19	46 (78.0)	21 (87.5)	0.38
Deprivation (Score EPICES ≥ 30)	38 (25.9)	20 (18.0)	18 (50.0)	**0.0001**	-	-	-
At least one parent with overweight (*n* = 122)	60 (49.2)	43 (46.7)	17 (56.7)	0.35	45 (48.4)	15 (51.7)	0.75
Mother overweight (*n* = 130)	28 (21.5)	13 (13.3)	15 (46.9)	**<0.0001**	16 (16.7)	12 (35.3)	**0.0232**
Father overweight (*n* = 123)	50 (40.7)	36 (39.1)	14 (45.2)	0.55	39 (41.9)	11 (36.7)	0.61
Study level of Mother: Higher level vs. lower to intermediate (*n* = 99)	63 (63.6)	54 (70.1)	9 (40.9)	**0.012**	57 (75.0)	6 (26.1)	**<0.0001**
Study level of Father: Higher level vs. lower to intermediate (*n* = 95)	57 (60.0)	47 (63.5)	10 (47.6)	0.19	49 (67.1)	8 (36.4)	**0.0098**

* Obese: BMI z-score ≥ 2.0. ** Deprivation: EPICES Score ≥ 30. *** Patient’s age was calculated at the time of the consultation in which parents completed the EPICES questionnaire. ^£^ NA Not appropriate due to small numbers: only 3 patients in Obesity group with an HQ Total Score and HQ-CT data. ^µ^ NA Not appropriate due to small numbers: only 5 patients in deprived group with an HQ Total Score and HQ-CT data.

**Table 2 jcm-11-02255-t002:** Characteristics of children with deprivation and their families classified by obesity. Variables are expressed in *n* (%) or in Med/Mean ± SD; for continuous variables (age, age at diagnosis, BMI Z-score, HQ, HQ-CT) *p* was calculated using Student or Wilcoxon test; for categorical variables (gender, genetic diagnosis, special schooling, parent’s obesity, study level of parents, treatment of GH/Psychotropic treatment, NIV, deprivation) *p* was calculated using Chi2 or Fisher exact test.

Variables	Non-Obese N = 20	Obese * N = 18	*p*
Boys	10 (50.0)	7 (38.9)	0.49
Age in years **	5.2 (7.1 ± 4.9)	10.1 (11.1 ± 5.0)	**0.021**
Age at diagnosis in months	0.7 (5.9 ± 14.7)	1.0 (9.6 ± 16.7)	**0.024**
Genetic subtype: Deletion (*n* = 38)	10 (50.0)	8 (44.4)	0.73
GH treatment	18 (90.0)	13 (72.2)	0.22
Psychotropic treatment	3 (15.0)	2 (11.1)	1.0
NIV	1 (5.0)	4 (22.2)	0.17
Special schooling (*n* = 24)	8 (80.0)	13 (92.9)	0.55
At least one parent with overweight (*n* = 29)	6 (42.9)	9 (60.0)	0.36
Mother overweight (*n* = 34)	4 (22.2)	8 (50.0)	0.091
Father overweight (*n* = 30)	4 (28.6)	7 (43.8)	0.39
Study level of Mother: Higher level vs. lower to intermediate (*n* = 23)	4 (30.8)	2 (20.0)	0.66
Study level of Father: Higher level vs. lower to intermediate (*n* = 22)	4 (33.3)	4 (40.0)	1.0

* Obese group: BMI z-score ≥ 2.0. ** Patient’s age was calculated at the time of the consultation in which parents completed the EPICES questionnaire.

**Table 3 jcm-11-02255-t003:** Results of multivariable logistic regression analysis of obesity in children with PWS.

*Logistic Regression*	Univariate Analysis		Multivariate Analysis (N = 130, 88.4%) 32 Obese (88.9% Among 36) 98 Non-Obese (88.3% Among 111)	
Variables	OR (95% CI)	*p*-Value	OR (95% CI)	*p*-Value
Boys	1.42 (0.66–3.02)	0.36		
Age in years ***	1.13 (1.05–1.22)	**0.0013**	1.16 (1.06–1.28)	0.002
Age at diagnosis ≤ 1 month	0.88 (0.39–1.98)	0.76		
Deprivation	4.55 (2.02–10.3)	**0.0003**	3.31 (1.26–8.73)	0.016
Genetic subtype: deletion (*n* = 146)	1.35 (0.63–2.89)	0.44		
Special schooling (*n* = 83)	2.42 (0.63–9.34)	**0.20**	*Special schooling not integrated in multivariate analysis due to data not applicable for children less than 6 years old*	
Mother overweight (*n* = 130)	5.77 (2.33–14.29)	**0.0002**	6.76 (2.36–19.37)	0.0004
Father overweight (*n* = 123)	1.28 (0.56–2.91)	0.56	*NS*	
Study level of mother: higher level vs. lower to intermediate (*n* = 99)	0.29 (0.11–0.79)	**0.015**	*Study level of parents not integrated in multivariate analysis due to too many missing data*	
Study level of father: higher level vs. lower to intermediate (*n* = 95)	0.52 (0.20–1.39)	**0.19**		
GH treatment	0.37 (0.13–1.07)	**0.065**	*NS*	
Psychotropic treatment	1.47 (0.47–4.55)	0.51		
NIV	3.24 (1.14–9.16)	**0.027**	*NS*	

*** patient’s age was calculated at the time of the consultation in which parents completed the EPICES questionnaire.

## Data Availability

The data presented in this study are available on request from the corresponding author.

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
