# Peer review of "Impact of Deprivation on Obesity in Children with PWS"

_jcm, 2022, doi:10.3390/jcm11082255_

Round 1
Reviewer 1 Report
The study conducted by Grolleau S et al. is interesting, well designed and written (among other things, the method chosen for ghrelin's dosage is to be appreciated). The study have identified also in French patients with PWS the socioeconomic background of the parents as a risk factor of developing obesity. These results are similar to data obtained by Slowetzky Amaro A et al. in Brazilian patients affected by PWS (doi.org/10.3389/fped.2022.746311). The different degree of development between France and Brazil highlights how the problem concerns the family of these patients regardless of the general context and as well discussed by the authors of the work, it requires changing the perspective of care by actively involving parents in nutritional therapeutic program.
I have only two minor observations:
The excessively light color of the graphs, as previous reported the authors discuss the possible daily therapeutic implications of their results, but considering the changes promoted by low-cost technological improvement (mobile phone-app) and by COVID restrictions it would be interesting to think about the e-health approach for these families.
Reviewer 2 Report
Title: Impact of deprivation on obesity in children with PWS
The study focuses on the problem of the prevalence of obesity among children suffering from Prader-Willi syndrome and the prevalence of low socioeconomic status of their family. The article is interesting and covers important aspects of the management of this genetic disorder. However, the article requires clarification on several points.
Please address the following issues:
- The authors use terms deprivation, social deprivation and deprived families, but it is not clear how they define the term deprivation. Please provide more explanation.
- The title of the article is not appropriate for its content, and the aims of the study specified in the abstract and introduction are not consistent. Please precise the aim or aims of the study.
- Please consider changing the title of the article and do not use abbreviations in the title (not PWS but full name of the disease).
- The methods section requires more explanation, including anthropometric measurement of children, the staff who provided the questionnaires, parents BMI (declaration of BMI or declaration of body mass and weight) as well as the ethics committee agreement. The details of EPICES score should be provided in the methodology section (not in the introduction).
- One of the main characteristics of PWS is severe obesity, and its prevalence varies according to age of patients. The study included children of different ages (from 3 months to 19 years). In my opinion, analysis of data in such a heterogeneous group of children is not appropriate. The group should be divided into infants, toddlers, preschoolers, school-aged children and adolescents (or another division into age groups), and all factors should be analyzed in these subgroups. The number of children in each subgroup should be reported.
- There is a lack of information on what obesity treatments were used in a group of children with PWS, especially among older children and adolescents. As nutrition is a very important part of the management of PWS, please explain whether the nutrition of children with PWS was assessed and whether it could have influenced the prevalence of obesity among children with PWS.
- The second column in tables 1 and 2 (all PW patients) presents the same results. I suggest to combine these two tables into one and to divide reported data into those related to children and parents. It seems that this method of presentation will make the results easier to understand.
- Similar in table 3 the data in the second column are the same as in the third column in table 2. Please consider another form of presentation of the results.
- The Hyperphagia Questionnaire was obtained only from 3 PWS obese patients. Is statistical analysis of such small numbers appropriate?
- In the discussion section, authors provide information that the age of the child is associated with obesity and this is explained by the known trajectory of the disease starting from difficulties in sucking and gaining weight to hyperphagia and obesity.
What about the age of parents and overweight and obesity problem? Was parental age associated with the prevalence of overweight or obesity?
- The authors reasonably point out that early identification of vulnerable families should be implemented as soon as possible after the diagnosis of PWS which is made in the first months of life in order to mitigate their impact on the occurrence of obesity. Please give your opinion about the model of the health care for PWS children and their family consisting of 1, 2 or 4 visits per year in the health care center. Is it an optimal model for heath care? Authors describe the care for PWS based on a national protocol as strengths of the study. Please provide more details of this protocol and indicate which study findings support this statement.
- The study limitations are not well described. Please revise all week points of your investigation from data collection to statistical analysis.
- Please explain which study findings are related to the statement in our study we could demonstrated that albeit prevalence of obesity is higher in children with PWS in deprived families, half of them benefit from the early multidisciplinary care and were not obese.
- Please explain the purpose of presenting results such as plasma total ghrelin or acetylated ghrelin, which were not included in the discussion. To my knowledge the authors have previously published the study demonstrating that hyperghrelinaemia starts early in life, prior to hyperphagia and obesity.
- Please also consider discussing the results regarding the prevalence and factors associated with obesity in relation to the general population. It seems that the reference should be the studies on this problem among PWS patients.
In conclusion, the article requires major revisions.
Reviewer 3 Report
The authors studied predictors of obesity in children with Prader-Willi syndrome, including socioeconomic status. Compared with the non-obese children, children with obesity had more deprived families This is an interesting report. Considering my personal experiences, I agree with these results. I have some questions.
- Compared with the non-obese children, children with obesity had more deprived families, older with a median of 10.1 vs 6.0 years. The difference of age is 4 years. Did the obese children show obesity or overweight when they were 6 years old?
- The mothers of obese children were more frequently obese and achieved less frequently high study levels. Is this phenomenon associated with some epigenetic factors?
- Deprivation in developing countries is often associated with poor income and shortage of food. Is overweight associated with ignorant in proper diet? Do you have special guidance for obese mothers by nutritionists in the clinic?
- Families at risk including social deprivation will require early identification and reinforced approach to prevent obesity. When the families live far from the national refence center, they may have disadvantage for visiting clinics. What do you think of the solution?
Round 2
Reviewer 2 Report
I appreciate the authors' efforts to improve the article. Although not all comments have been completely explained, I accept the current form of the article.